# Positive Regulation of Cellular Proteins by Influenza Virus for Productive Infection

**DOI:** 10.3390/ijms26083584

**Published:** 2025-04-10

**Authors:** Jiayu Cong, Ting Wang, Bumsuk Hahm, Chuan Xia

**Affiliations:** 1Department of Pathogen Biology and Microecology, College of Basic Medical Sciences, Dalian Medical University, Dalian 116044, China; tangwei0709@hotmail.com; 2Department of Bioengineering, College of Life Science and Technology, Jinan University, Guangzhou 510632, China; wangting1988@jnu.edu.cn; 3Departments of Surgery & Molecular Microbiology and Immunology, University of Missouri, Columbia, MO 65212, USA

**Keywords:** influenza virus, virus–host interaction, virus life cycle, immune evasion, signaling pathways

## Abstract

Influenza viruses cause annual epidemics and occasional pandemics through respiratory tract infections, giving rise to substantial morbidity and mortality worldwide. Influenza viruses extensively interact with host cellular proteins and exploit a variety of cellular pathways to accomplish their infection cycle. Some of the cellular proteins that display negative effects on the virus are degraded by the virus. However, there are also various proteins upregulated by influenza at the expression and/or activation levels. It has been well-established that a large number of host antiviral proteins such as type I interferon-stimulated genes are elevated by viral infection. On the other hand, there are also many cellular proteins that are induced directly by the virus, which are considered as pro-viral factors and often indispensable for rigorous viral propagation or pathogenicity. Here, we review the recent advances in our understanding of the cellular factors deemed to be upregulated and utilized by the influenza virus. The focus is placed on the functions of these pro-viral proteins and the mechanisms associated with promoting viral amplification, evading host immunity, or enhancing viral pathogenicity. Investigating the process of how influenza viruses hijack cellular proteins could provide a framework for inventing the host-factor-targeted drugs to conquer influenza.

## 1. Introduction

Influenza viruses cause seasonal and pandemic influenza with a significant morbidity and mortality and are considered a threat to animal and human populations all over the world [1]. Influenza viruses belong to the Orthomyxoviridae family and comprise enveloped viruses with a negative-sense single-strand RNA (ssRNA) genome that can be categorized into types A, B, C, and D [2,3,4,5]. The influenza viral genome is split into eight segments, encoding at least eleven proteins. The surface of the viral envelope is decorated with viral hemagglutinin (HA), neuraminidase (NA), and matrix protein 2 (M2). The matrix 1 (M1) protein is underneath the envelope, providing support. The viral ribonucleoprotein (vRNP) complex contains viral polymerase subunits PB1, PB2, and PA, as well as viral RNA (vRNA), which is coated by oligomers of viral nucleoprotein (NP). There is also viral non-structural protein 1 (NS1), which is well-established as an important interferon antagonist [6,7,8,9], NS2, which is involved in the nuclear export of vRNPs, and PB1-F2 protein, which is a second gene product of PB1 and has multiple functions during infection [9,10,11].

Like other acellular organisms, influenza viruses are unable to self-replicate and have to subvert a host cell to propagate. To obtain a suitable environment to effectively replicate and spread, influenza viruses always hijack the host cells’ mechanisms. Viruses manipulate or utilize the cellular proteins, making them support each step of the viral life cycle. However, hosts have also developed various antiviral strategies to restrict these viruses. Therefore, the interaction between influenza viruses and their hosts is extremely complicated.

Many lines of research have revealed the inhibition of cellular protein expression by influenza viruses. These proteins are either reduced at the mRNA level or directly degraded by the virus [12,13,14]. Earlier, we reviewed the influenza-virus-induced degradation of cellular proteins, highlighting the modes of how viruses co-opt the host by degrading these proteins to support viral propagation [10]. However, there are few articles that summarize the cellular proteins upregulated by influenza viral components during infection. To our knowledge, influenza viruses positively regulate a considerable number of cellular proteins. Qiao et al. analyzed the protein expression differences between uninfected and infected MDCK cells with influenza A virus (IAV), using DIA proteomics analysis technology. A total of 266 differentially expressed proteins were identified, 157 of which were upregulated, while 109 were downregulated [15]. As expected, viral infections activate the production of type I interferons (IFNs), which bind to the receptors and trigger the expression of a large group of host genes that exert antiviral effects [16,17,18]. On the other hand, during infection, viral components are capable of specifically increasing a series of host proteins through different mechanisms. These proteins are non-interferon-induced and are typically essential for virus propagation. They either have increased expression levels or enhanced activities (mainly enzymes) during infection with the influenza virus. The upregulation of cellular proteins upon infection is involved in numerous signaling pathways and is considered a consequence of the utilization of these proteins by influenza viruses.

In this review, we discuss the influenza-virus-induced increased expression and activation of cellular proteins, highlighting the modes of how these viruses facilitate self-propagation by utilizing these proteins.

## 2. Influenza Virus Life Cycle

To better understand the interplay between influenza viruses and their host, we first look into the life cycle of this virus. The whole cycle can be divided into several stages as follows: attachment to the host cell; penetration into the cell; vRNP transport into the nucleus; synthesis of the viral genome and protein; assembly of the virion; and budding from the plasma membrane of the host cell (Figure 1), which have been extensively investigated and reviewed [19,20,21,22,23,24].

Upon influenza virus infection, viral HA first binds to the sialic acid linked to galactose on the surface of host cells’ membrane [25,26,27]. After that, the virus penetrates the cell through receptor-mediated endocytosis. A low pH value triggers the fusion of the endosomal membrane with the viral envelope, releasing vRNPs into the cytoplasm. This procedure needs the participation of viral ion channel protein M2 [28]. The influenza virus performs replication in host cells’ nuclei, which is different from many other RNA viruses. Viral NP, PA, PB1, and PB2 are capable of binding to the cellular nuclear import machinery, helping vRNP enter the nucleus [29]. The virus hijacks the host’s transcriptional machinery for its own benefit in many ways [30,31]. Segments 1~6 of the viral genome encode for six proteins individually, while segment 7 encodes for M1 and M2 and segment 8 encodes for NS1 and NS2 due to differential splicing. There is another nonstructural protein encoded by the PB1 gene segment from an alternative ORF called PB1-F2, which has multiple functions in viral pathogenicity. After a complex synthesis and assembly process, the newly synthesized vRNPs are exported from the nucleus to the cytoplasm with the help of NP, M1, and NS2 [29,32,33]. After vRNP export, the newly synthesized viral components are packaged into progeny virions and released from the cell. As an enveloped virus, influenza virus also forms its envelope by utilizing the host cell’s plasma membrane, which contains viral HA, NA, and M2 on the surface. At last, viral NA cleaves the sialic acid residues from glycoproteins and glycolipids, helping the viral particles release from the host. The whole process of influenza virus replication cycle is very complicated, and each step is inseparable from the interaction between the virus and host cellular proteins.

## 3. Influenza-Virus-Induced Upregulation of Cellular Proteins

During infection, a number of cellular proteins are required for the replication of influenza viruses, and some of them are markedly upregulated at the gene and/or protein level by the virus to support viral propagation more efficiently. The following contents are classified based on the regulatory functions of those cellular proteins in different biological processes.

### 3.1. Molecules Supporting the Virus Life Cycle

#### 3.1.1. Viral Entry, Uncoating, and vRNP Import

As acellular organisms, influenza viruses require host cellular materials as well as mechanisms to execute their life activities. The virus life cycle demands the support of diverse host factors (Figure 1). At the early stage of influenza virus infection, the p38 mitogen-activated protein kinase (MAPK) was shown to be activated at the site where the virus attaches to the cell. The phosphorylated p38 MAPK then plays a determinant role in mediating virus entry in a TLR4- and MyD88-dependent way [34]. Su et al. conducted an shRNA-based screen and identified a cellular protein, Itch, as an important regulator for entry at the uncoating stage of influenza virus infection. Itch was phosphorylated and recruited to the virion containing endosomes and was proven to act in the release of vRNP from there. Additionally, Itch was shown to interact and ubiquitinate viral M1 protein [35]. The vacuolar (H+)-ATPases (V-ATPases) facilitate the release of vRNP into the cytoplasm by acidifying the endosomal interior [36]. The activity of V-ATPase was shown to be elevated during IAV infection, which was mediated by extracellular signal-regulated kinase (ERK) and phosphatidylinositol 3-kinase (PI3K). The inhibition of these two kinases reduced V-ATPase activity and the acidification of intracellular compartments in infected cells [37]. Protein kinase C (PKC) was proven to be activated by viral HA during infection, thus regulating vRNP import into the nucleus [38,39]. Interferon-induced transmembrane protein 3 (IFITM3) is highly induced by both type I and type II interferons upon influenza virus infection [40,41], which displays antiviral activity by inhibiting viral entry through the blockade of viral membrane fusion to the endosomal membrane [42]. It has been reported that IAV infection increases the monomethylation of IFITM3 at its 88 lysine residue (IFITM3-K88me1), therefore reducing the antiviral activity of this protein [43]. Recently, Yang et al. reported galectin-3, a member of the β-galactoside-binding animal lectin family, was increased in the bronchoalveolar lavage fluid and lungs of IAV-infected mice, and the level of galectin-3 was correlated with viral load. Importantly, the expression of galectin-3 increased vRNP nuclear import, viral transcription, and replication. This effect was further shown to associate with the interaction of galectin-3 with the viral PA subunit [44].

#### 3.1.2. Viral Replication, Transcription, and Translation

Influenza viral transcription and replication occur in the nuclei of host cells and are mediated by viral RNA-dependent RNA polymerase (RdRp), which consists of PB1, PB2, and PA subunits. Many host proteins are recruited to help carry out these processes. Jorba et al. characterized the binding partners of influenza viral RdRp, and a series of cellular proteins were identified [45]. A follow-up study revealed one of those proteins, NXP2/MORC3, is increased during influenza virus infection [46]. NXP2/MORC3 belongs to the Microrchidia family and is associated with the nuclear matrix, displaying RNA-binding activity [47,48]. The shRNA-based knockdown of NXP2/MORC3 markedly reduced viral propagation. Further investigation determined that this protein is important for the transcription step of the virus life cycle [46]. A yeast two-hybrid screen revealed another cellular protein interacting with influenza viral RdRp, which is pyruvate kinase M2 (PKM2) [49]. Miyake et al. determined that the C-terminus of PKM2 interacts with the C-terminal region of the PA subunit, and its protein level is upregulated during IAV infection. Further study indicated that PKM2 is crucial for influenza virus replication by transferring a phosphate group to PA, therefore converting the function of viral RNA polymerase from transcriptase to replicase [49].

The mechanistic target of rapamycin (mTOR) promotes protein synthesis by activating downstream effectors. The expression of mTOR was shown to increase in A549 cells at both the mRNA and protein level up to 24 h post-infection (hpi) with IAV but decrease at 48 hpi [50]. Additionally, viral NP was proven to interact with mTOR and regulate the dynamic expression of mTOR. Furthermore, the decrease in mTOR levels at the late timepoint of infection was determined to be caused by a miRNA (miR-101), which targeted the mRNA of mTOR to reduce its expression [50]. Coincidentally, another research has shown that IAV NP and M2 induce autophagy via the AKT-mTOR pathway [51]. NP and M2 mediate the induction of autophagy by upregulating the protein levels of HSP90AA1, which interacts with PB2 and results in an increase in viral RNA synthesis [51]. Kuss-Duerkop et al. have shown that AKT was phosphorylated/activated during influenza virus infection by two different ways. PDPK1-mediated phosphorylation of AKT is required for mTORC1 activation by the virus, while viral NS1-promoted phosphorylation of AKT regulates apoptosis [52]. Several research studies have documented that the influenza virus increases both the levels and activation status of a 58 kDa cellular inhibitor of PKR (P58IPK) [53,54,55,56]. Mechanistically, viral NP interacts with Hsp40 and induces the dissociation of Hsp40 from P58IPK, therefore activating this protein. The activated P58IPK then suppresses PKR as well as elF2α, resulting in the enhanced translation of influenza viral mRNA. The splicing of viral RNA is essential for efficient translation into viral proteins. Cellular serine and arginine-rich splicing factor 5 (SRSF5) was shown to be inducted by IAV infection, then binds to viral M mRNA as well as U1 small nuclear ribonucleoprotein (snRNP), promoting M mRNA splicing and M2 generation [57]. Protein acylation is a post-translation modification that has a beneficial effect for influenza virus replication in vivo, and at least two viral proteins (HA and M2) need to be S-acylated during infection [58,59]. Influenza viral NS1 was proven to upregulate ZDHHC22 at the mRNA level, which belongs to the ZDHHC acyltransferase family [60]. However, ZDHHC22 knockout did not alter the production of infectious particles of IAV, suggesting the physiological role of the NS1-mediated elevation of ZDHHC22 remains unclear [60].

#### 3.1.3. vRNP Assembly and Nuclear Export

Yang et al. identified the host factors that interact with influenza vRNP. They showed that LYAR, a cellular nucleolar protein, interacts with vRNP and is upregulated during IAV infection. LYAR was further proven to enhance vRNP assembly, thus facilitating viral RNA synthesis [61]. Ren et al. investigated the global expression profiling of swine-encoded genes in response to swine H1N1/2009 IAV in newborn pig trachea cells [62]. Among the total of 166 genes that were differentially expressed, polo-like kinase 3 (PLK3) was further investigated. The mRNA and protein levels of PLK3 were confirmed to be upregulated during viral infection. Additionally, PLK3 was determined to interact with viral NP and markedly increased the phosphorylation and oligomerization of NP, promoting the vRNP assembly [62]. The vRNP nuclear export is dependent on CRM-1 and its cofactor Ran-binding protein 3 (RanBP3) [63,64]. It has been reported that RanBP3 was phosphorylated at Ser58 during the infection of IAV. This phosphorylation is regulated by both PI3K/AKT and Ras/ERK/RSK pathways and is important for mediating vRNP nuclear export [64,65]. The accumulation of viral HA in the cellular membrane induces Raf activation [66]. The activated Ras/ERK/RSK signaling then promotes the nuclear export of vRNP through inducing the phosphorylation of viral NP at Ser269 and Ser392 [67]. The inhibition of Raf signaling results in the nuclear retention of vRNP [38,67]. Seo et al. demonstrated that IAV infection increases the expression and activation of sphingosine kinase 1 (SphK1), which in turn regulates diverse cellular signaling pathways. The inhibition of SphK1 suppressed virus-induced NF-κB activation, reducing the synthesis of viral proteins and RNAs. Additionally, SphK1 blockade interfered with the activation of RanBP3, inhibiting the CRM1-mediated nuclear export of vRNPs [68]. Another cellular factor elevated by IAV that facilitates viral vRNP nuclear export is chaperonin containing TCP1 subunit 5 (CCT5). CCT5 was proven to interact with viral NP, PB1, and PB2, but not with PA, playing an essential role in vRNP nuclear export [69].

#### 3.1.4. Package of Progeny Virus

Many lines of evidence have implied that influenza viral RNA polymerase is associated with host RNA polymerase II (Pol II) and triggers the degradation of Pol II, acting as a cellular shut-off [12,70,71,72,73]. Interestingly, one cellular protein, hCLE/C14orf166, was shown to interact with influenza viral polymerase as well as host Pol II, stimulating the activities of both proteins [74]. Moreover, the protein levels of hCLE increased during infection in a viral-replication-dependent manner. hCLE was shown to interact with vRNP in the nucleus as well as in the cytoplasm at a late stage during infection, incorporating into the viral particles [74]. The synthesis of viral glycoproteins generates burden in the endoplasmic reticulum (ER), triggering the unfolded protein response (UPR) [75,76]. Recently, Marques et al. revealed that IAV infection activates the inositol requiring enzyme 1 (IRE1), which mediates one of the UPR pathways, resulting in the accumulation of virus-induced insoluble protein aggregates. This accumulation was further proven to favor the package stage of the virus life cycle [77].

### 3.2. Host Factors Regulating Viral Immune Evasion

#### 3.2.1. Suppression of Innate Immune Response

It has been well established that upon influenza virus infection the cellular sensors such as RIG-I sense the presence of viral RNA, thus initiating the production of type I IFNs. The IFNs bind to the receptors and trigger the expression of numerous interferon-stimulated genes (ISGs), mainly exerting antiviral function [16]. A recent article has reviewed the harmful effects on the host of type I IFN responses [78]. Besides type I IFNs, there is type III IFN signaling, which is also proven to display antiviral activity [79]. Influenza viruses vigorously evade a host’s antiviral response utilizing diverse strategies [80]. In addition to suppressing or directly inducing the degradation of regulators within the type I IFN signaling pathway, the influenza virus also elevates certain cellular proteins to help antagonize this powerful antiviral response (Figure 2). The highly pathogenic H5N1 IAV induces the enhanced expression of FAT10, which is proven to positively regulate viral replication. Further, the upregulation of FAT10 is triggered by viral RNA and is mediated by RIG-I and NK-κB. Importantly, FAT10 upregulation facilitates virus replication by inhibiting type I IFN signaling [81]. Pauli et al. have revealed that the mRNA levels of the suppressor of cytokine signaling-3 (SOCS-3) but not SOCS-1 increase during IAV infection. The upregulation of SOCS-3 mRNA levels was proven to be dependent on NK-κB early in the viral replication cycle. Furthermore, the viral induction of SOCS-3 appears to be involved in the suppression of the antiviral response, since the viral propagation was reduced in SOCS-3-deficient cells. SOCS-3 knockout resulted in the constitutive activation of STAT1 and enhanced expression of ISGs [82]. Nevertheless, another group have demonstrated that IAV infection induces the robust expression of SOCS-1, leading to the inhibition of STAT1 activation, therefore disrupting IFN-λ antiviral signaling. The authors also showed that the SOCS-1 expression resulted in NF-κB activation, thereby enhancing IFN-λ molecule expression. The excessive production of IFN-λ somehow impaired the antiviral response [83]. SOCS-1 was also shown to mediate the ubiquitination and degradation of JAK1, resulting in the inhibition of both type I and type II IFN responses [84]. IAV can modulate IFN-λ signaling by inducing the expression of a certain E3 ubiquitin ligase. For example, it has been suggested that influenza virus infection increases the level of an E3 ligase subunit, FBXO45, which in turn induces the ubiquitination and degradation of IFN-λ receptor subunit IFNLR1 [85]. IAV was also reported to attenuate interferon signaling transduction by suppressing the expression of IRF3 and STAT1 during infection. Hu et al. determined that IAV infection induces the expression of transcription factor RUNX1. The overexpression of this protein efficiently enhances the production of progeny viruses. They further revealed that RUNX1 is a negative regulator for type I IFN, which attenuates the signaling transduction by hampering the expression of IRF3 and STAT1 during IAV infection [86].

Besides SphK1, the other isoform of SphK, SphK2, was also proven to be manipulated by IAV and displays pro-viral activity [87,88]. Xia et al. demonstrated that IAV infection led to an increased expression and phosphorylation of SphK2 in host cells. Furthermore, the pharmacologic inhibition of SphK2 attenuated IAV replication and substantially improved the viability of mice following IAV infection [87]. Follow-up research determined that SphK2 interacts with the IFN-β promoter through the binding of demethylase TET3. The interaction recruits HDAC1 to the promoter and enhances its deacetylation, therefore leading to the inhibition of IFN-β transcription [89]. IAV is also demonstrated to suppress the interferon-mediated antiviral response by activating the epidermal growth factor receptor (EGFR) [90]. The phosphorylation of EGFR and ERK was detected at the early stage of IAV infection, while the inhibition of either protein resulted in the increased expression of type I and type III IFNs with reduced viral replication. Further, the cellular Src homology region 2-containing protein tyrosine phosphatase 2 (SHP2) was proven to play a crucial role in the IAV-induced activation of EGFP/ERK [91]. The highly pathogenic avian influenza H5N1 virus was reported to upregulate both the mRNA and protein levels of host N-myc downstream-regulated gene 1 (NDRG1) by viral M1 and PB1. Overexpressed NDRG1 then suppressed the IKKβ-mediated production of type I IFNs in an NK-κB-dependent fashion, therefore, facilitating viral replication [92].

The nonclassical human leukocyte antigen (HLA) class I molecule, also known as the human major histocompatibility complex class I (MHC-I), is recognized by the killer cell immunoglobulin-like receptors on NK cells [93,94]. HLA-G is well established as a tolerogenic molecule suppressing both innate and adaptive immunity [95,96]. LeBouder et al. analyzed the expression of HLA-G in lung epithelial cells following influenza virus infection. HLA-G was shown to be elevated at both the mRNA and protein levels in a strain-dependent manner [97]. However, the mechanisms by which HLA-G is induced during infection remains unknown. Mahmoud et al. have demonstrated that IAV infection in mice is associated with the increased expression of mouse MHC-I on lung epithelial cells [98]. Wang et al. found that the increased expression of MHC-I was mediated by ERAP1 expression caused by NS1 during IAV infection [99]. Specifically, Rahim et al. observed that infection with IAV (A/Fort Monmouth/1/1947(H1N1)) in lung epithelial cells significantly increased the expression of HLA-B, -C, and -E that bind to the inhibitory receptors of NK cells [100]. Moreover, aberrant internally deleted viral RNAs (mini viral RNAs) and defective interfering RNAs expressed from an IAV mini-replicon were shown to be sufficient for inducing HLA upregulation. The authors further evidenced that MAVS was required for HLA upregulation in response to IAV. The HLA upregulation was suggested to be advantageous to the virus by helping it evade host innate immunity [100].

#### 3.2.2. Regulating PD-1:PD-L1 Signaling

Although influenza viruses preferentially replicate in respiratory epithelial cells, other cell types such as monocytes, macrophages, and T lymphocytes are also susceptible to IAV infection [101,102,103,104]. Programmed death 1 (PD-1):PD-ligand 1 (PD-L1) signaling displays a critical role in regulating T cell response during chronic infections or cancer. Many studies have revealed the IAV-mediated alteration of PD-1 and/or PD-L1 expression in different cell types, as well as the importance of these molecules in the clearance of influenza virus [105,106,107,108,109] (Figure 3). Valero-Pacheco et al. demonstrated that influenza A (H1N1) pdm09 virus induces the expression of both PD-L1 and PD-1 on human dendritic cells and T cells. PD-L1 expression was proven to impair the T cell response against IAV by promoting the cell death of CD8+ T cells and reducing cytokine production [106,110]. This phenomenon was further confirmed by clinical data from the first and second 2009 flu pandemics in Mexico City [106]. Specifically, highly pathogenic IAV induces more PD-L1 expression on virus-specific CD8+ T cells [111]. Consistently, other studies have revealed that the blockade of the PD1:PD-L1 signaling reduces the virus titer in lung tissue and alleviates the symptoms during influenza virus infection [105,112]. Zhang et al. investigated the role of PD-L1 during the infection of pulmonary microvascular endothelial cells (RPMECs) with the H9N2 virus. PD-L1 was upregulated at both the mRNA and protein levels upon infection, and the virus-induced PD-L1 expression markedly reduced the production of IL-2, IFN-γ, granzyme B, and perforin from T cells [108]. There is evidence showing that invariant natural killer T (iNKT) cells are important for defending the host against IAV invasion, while the expression of both PD-1 and PD-L1 is elevated upon IAV infection on iNKT cells. A lack of PD-L1 expression was shown to improve the course of infection [113]. PD-L1 was also shown to be strongly upregulated in A549 cells in response to IAV infection. Exogenous overexpression of PD-L1 promoted viral replication. Further, PD-L1 was found to interact with cellular protein SHP2, and the overexpression of PD-L1 decreased the phosphorylation of SHP2 and ERK, which was induced by IAV infection. PD-L1 overexpression was further proven to reduce the expression of type I and type III IFNs as well as ISGs in A549 cells [114].

### 3.3. Molecules Contributing to Viral Pathogenicity

The up-regulation of cellular proteins by influenza viruses can help regulate the infectivity and exert their pathogenicity of the viruses, including the regulation of inflammatory response during infection (Figure 4). To understand the pathology and immunological response to viral HA from H7N9 avian IAV, Zhang et al. analyzed the expression of proinflammatory factors and miRNA in THP-1 cells by treating cells with H7N9 HA [115]. HA protein significantly increased the expression of several cytokines such as IL-1α, IL-1β, and IL-6. Notably, HA increased let-7e expression in THP-1 cells and decreased let-7e levels in the medium. Further study revealed that this HA-mediated upregulation is dependent on toll-like receptor 4 (TLR4) signaling and NF-κB [115]. Recently, Zhao et al. analyzed the regulatory function of HA alone on the expression of IL-6 and intercellular adhesion molecule 1 (ICAM-1), which play important roles in the pathological and inflammatory response upon infection with viruses such as influenza and SARS-CoV-2 [116,117,118,119]. The incubation of human umbilical vein endothelial cells (HUVECs) with HA of H1N1 markedly increased the mRNA and protein levels of ICAM-1 and IL-6. A similar effect was observed in the lung tissues of infected mice, suggesting a new pathogenic function of HA during influenza virus infection [116].

Influenza virus was proven to upregulate the cellular signaling with-no-lysine-kinase-4 (WNK4) pathway [120]. WNK4 was reported to inhibit epithelial sodium channels (ENaCs) that are important for water and salt transport in the respiratory system [121,122,123]. Hou et al. have shown that WNK4 was increased at protein levels during influenza virus infection, which resulted in a virus-induced reduction in ENaC activity, thus impairing fluid transport in the airway [120]. Dlugolenski et al. investigated the influenza virus reassortment rate in swine and human cells. They demonstrated that infection with recombinant human H1N1 barbering swine NA (rhuH1N1-swNA) or PA (rhuH1N1-swPA) triggers significant MIP-2 expression [124]. Since chemoattractant MIP-2 mediates neutrophil recruitment [125], the overexpression of MIP-2 induced by recombined NA and PA resulted in substantial pulmonary neutrophilia. However, MIP-2 was not increased upon infection of the parental viruses, suggesting the potential for the reassortant to have increased pathogenicity linked to the swine NA and PA genes that are related to MIP-2 expression [124].

Acute influenza infection has been reported to associate with neurological symptoms such as influenza-associated encephalopathy (IAE) [126,127,128]. Ding et al. investigated the immune response and differential protein profiles during IAV infection in BV2 MGs cells. The results indicated that the expression of osteoponin (OPN) was upregulated at 16 and 32 hpi, leading to aggravated brain damage and inflammation [129]. The extracellular cleavage of influenza viral HA (HA0) by host trypsin-like protease is essential for the infectivity of IAV [19,20]. Lea et al. reported that IAV infection significantly upregulates ectopic trypsin in brain endothelial cells, which may potentiate brain vascular hyperpermeability and tissue damage [130]. Follow-up studies have clarified the pathological roles of upregulated trypsin in the progression of myocarditis in severe influenza [131,132]. Clinically, influenza virus infection can cause acute cardiovascular events [133,134]. Lee et al. examined the expression levels of matrix metalloproteinases (MMPs) during the IAV infection of human cells. MMP-13 was shown to increase during IAV infection, and the expression of MMP-13 was regulated through the p38 MAPK signaling pathway. Further research showed that collagen was degraded with MMP-13 expression in the atherosclerotic plaque lesion, leading to destabilization of vulnerable atherosclerotic plaques in the artery [135]. IAV also elevates the level of MMP-9, which contributes to severe tissue damage [136,137]. In addition, IAV infection elevates the expression of eotaxin, a type of CC chemokine, which may participate in the pathogenesis of airway inflammatory disease caused by influenza [138]. Tian et al. investigated the H3N2 IAV infection during chronic rhinosinusitis with nasal polyps (CRSwNP). The data showed that IAV increases the expression of oncostatin M (OSM), which is implicated in CRSwNP as a possible mechanism of tight junction impairment [139].

Influenza virus cross-species transmission is restricted by the host. However, viruses overcome this restriction by accumulating mutations that allow them to adapt to a new host [140]. A number of cellular factors have been identified to be involved in influenza virus host adaptation [141,142,143]. Liu et al. found that histone H1.2 (encoded by HIST1H1C) regulates human and avian influenza virus replication in different ways. They demonstrated that the expression, phosphorylation, and methylation levels of HIST1H1C are decreased when infected with H1N1 influenza virus but are increased when infected with the H5N1 virus. Consistently, HIST1H1C was shown to positively regulate the H5N1 virus infection, whereas it negatively regulates H1N1 virus replication. PB2 was identified as the key factor that alters HIST1H1C levels. Moreover, Sp1 was proven to be crucial in regulating HIST1H1C expression by PB2 [144]. The influenza viral regulation of gene expression also relates to the host species. Taye et al. compared the host gene expression signatures between cell lines from three species (human, chicken, and canine) in response to six different influenza A viruses. OSBPL1A and ARHGAP21 were highly expressed in chicken cells, while CCL5 was highly expressed in MDCK and A549 cells [145], suggesting the different pathogenic roles of these proteins in different host species. 

Influenza viruses have been reported to play important roles in chronic diseases [146,147,148]. Particularly, influenza virus infections can cause severe complications in human immunodeficiency virus type-1 (HIV-1)-infected individuals. Previous research revealed that the influenza virus infection of HeLa-CD4+ cells resulted in increasing levels of CXCR4 transcripts, as well as the surface expression of this receptor, suggesting a possible role of influenza virus in contributing to AIDS progression by modulating coreceptor availability [149]. Sun et al. assessed the influenza viral modulation of the HIV-1 long terminal repeat (LTR) in human CD4+ T cells and found that the virus was capable of promoting the expression of the HIV-1 LTR-driven reporter gene. Furthermore, the positive action of influenza virus on HIV-1 LTR activity was shown to be mediated through the induction of NF-κB [150].

### 3.4. Proteins Triggering Pro-Viral Cellular Signaling

Influenza viruses employ multiple strategies to manipulate host cellular signaling. Besides the signaling directly involved in the viral life cycle and immune evasion that have been detailed in the previous sections, there are also other important cellular signaling pathways (Figure 4) that are induced by the virus for efficient influenza viral propagation. Unlike many viruses that downregulate p53 [151,152,153,154], IAV infection activates the p53 pathway. For instance, viral NS1 was proven to regulate p53-mediated apoptosis at a cellular level. Yan et al. have shown that the expression of NS1 from the H7N9 strain activated caspase 3/7 and increased the protein levels of cleaved caspase 7 as well as poly (ADP-ribose) polymerase (PARP) in A549 cells, consequently increasing the activation of p53 and inducing apoptosis [155]. The H7N9 virus can be detected in brain tissues and associated with the central nervous system with symptoms in infected animals [156], which is at least partially mediated by viral NS1. The same group (Yan et al.) determined that NS1 from the H7N9 strain increases the expression of inducible nitric oxide synthase (iNOS) and increases NO release in Neuro2a cells, inducing cell growth arrest and cellular senescence [157]. They further found that this regulatory effect is dependent on p38 MAPK signaling [157].

PI3K/AKT signaling is another pathway that influences IAV infection. It has been established that IAV activates the PI3K/AKT pathway in order to delay virus-induced apoptosis, thus providing sufficient time for virus replication [158,159,160]. Shin et al. have shown that AKT phosphorylation was elevated in a PI3K-dependent way during the IAV infection of A549 cells [161]. Further, IAV NS1 was shown to bind to the p85β subunit of PI3K and activated this pathway [6,162]. Li et al. further reported that NS1 binds to the iSH2 domain of p85β and markedly increases p85β-associated PI3K activity. Further study evidenced that NS1, p85β, and p110 form a complex that facilities the regulatory function of NS1 [163]. c-Jun terminal kinase (JNK) is a stress-activated protein kinase that regulates autophagy. It has been reported that AKT activation in the PI3K pathway suppresses JNK [164]. Interestingly, the H1N1 virus activates the PI3K pathway but does not activate JNK, whereas the H5N1 virus is incapable of activating the PI3K pathway but activates JNK; on the other hand, the H9N2 virus activates both pathways [165]. The H5N1 virus-specific activation of JNK was likely through the virus-induced activation of TGF-β-activated kinase 1 (TAK1) [166]. JNK was further shown to play a critical role in regulating vRNA and protein synthesis during infection with the H5N1 virus [167]. Viral NS1 was suggested to mediate JNK activation and apoptosis independently [168].

Influenza virus was proven to alter the transforming growth factor beta (TGF-β) signaling pathway. The levels of TGF-β were shown to increase in mice infected with influenza virus. Viral NA was further determined to directly activate TGF-β and induce apoptosis in cells [169,170]. Pneumonia caused by bacterial co-infection with influenza virus is the leading cause of mortality during flu pandemics [171], and TGF-β has been proven to mediate the viral–bacterial co-infections. Li et al. have determined that influenza virus infection activates TGF-β, which in turn upregulates the expression of adhesins, resulting in increased host susceptibility to bacterial co-infections such as Streptococcus pneumoniae [172]. Recent research has demonstrated that the H9N2 avian influenza virus enhances the avian pathogenicity *Escherichia coli* (APEC) infection of chicken oviduct epithelial cells (COECs), causing chicken salpingitis. Mechanism studies have revealed that NS1 of the H9N2 virus activates TGF-β signaling, thus upregulating the expression of fibronectin, which promotes APEC adhesion onto COECs [173]. Another independent study conducted by Wang et al. also revealed the contribution of elevated TGF-β1 to APEC adhesion after H9N2 infection with consistent results [174]. Zhu et al. found that IAV infection induces the expression of collagen triple helix repeat containing 1 (CTHRC1), which is a novel secreted glycoprotein displaying diverse functions such as regulating the generation of TGF-β signaling and reparation of vascular intimal injury. Clinical studies revealed a higher expression of CTHRC1 in patients infected with IAV in comparison to healthy people. Further research demonstrated that IAV upregulated CTHRC1 through viral NS1. NS1 specifically enhances the gene promoter activity of CTHRC1, thus increasing the level of this protein [175].

### 3.5. Host Factors Benefiting Influenza Virus Through As-Yet-Undefined Mechanisms

In addition to the lists summarized above, there are other cellular proteins that are upregulated during influenza virus infection via as-yet-unknown mechanisms. These proteins participate in many cellular processes and mostly display positive impacts on influenza viruses. By performing a ddRT-PCR, Yang et al. determined several cellular genes that were upregulated during influenza virus infection. They confirmed the increased expression of one gene-PRPF8 by RT-PCR. The overexpression of PRPF8 enhanced viral replication, while the knockdown of this gene significantly reduced viral production, indicating a positive role of PRPF8 in viral infection. Further investigation revealed that the expression of viral NS1 or PB1 triggered the elevation of PRPF8 levels [176]. However, how PRPF8 contributes to influenza virus replication needs further investigation. It was reported that a key amino acid at 627 in viral PB2 from highly pathogenic H5N1, H7N9, and H10N8 isolated from China had mutated to K under natural conditions. To investigate the mechanism of the increased pathogenicity of these viruses, Qi et al. analyzed the differential expression of proteins in mouse lungs in response to PB2-K627E IAV infection [177]. Five proteins were upregulated and nine were downregulated at 12 hpi, while ten were upregulated and 25 downregulated at 72 hpi. These proteins are involved in many biological processes such as cytoskeleton remodeling, the regulation of glucocorticoids, signaling transduction, and inflammation. Further, three upregulated proteins (moesin, ezrin, and sp-A) were confirmed in A549 cells [177].

PARPs are a family of proteins that catalyze the enzymatic activity of the transferring of ADP-ribose motifs to target proteins [178,179,180]. Several of the PARP family members have been shown to display regulatory activities on different viruses, including the influenza virus [181,182,183]. Bamunuarachchi et al. have shown that both the mRNA and protein levels of PARP5b (tankyrase 2, TNKS2) were increased upon infection with different IAV strains. The upregulation of TNKS2 was also observed in the lung tissues of IAV-infected mice. They further demonstrated TNKS2 was targeted by a cellular microRNA (miR-206) that displays an anti-influenza function by inhibiting TNKS2 [184]. A follow-up study from the same group revealed that, similar to TNKS2, the TNKS1 isoform was also elevated by influenza virus and targeted by a miRNA, miR-9-1 [185]. However, the precise roles of TNKS in influenza virus infection were not determined [185].

### 3.6. Influenza Viral Upregulation of ncRNAs

IAV infection also affects the expression patterns of a large number of cellular non-coding RNAs (ncRNAs), including microRNA (miRNA), small interfering RNA (siRNA), long non-coding RNA (lncRNA), and so on [186,187,188,189,190,191]. Cellular miRNA is a class of non-coding single-stranded RNA molecules that are involved in the regulation of gene expression. Influenza viruses are known to alter the expression of miRNAs, thus modifying the cellular environment to facilitate viral propagation, which has been reviewed extensively by many authors [192,193,194,195,196]. For example, Dong et al. have shown that IAV significantly induced the expression of miR-9 that enhanced viral gene expression and the production of progeny virion [197]. Jiao et al. investigated the interaction between NS1 and miRNA in host cells, revealing a total of four miRNAs that were upregulated under the expression of NS1, while two miRNAs were downregulated [198]. Human airway epithelial cells exposed to IAV induced the expression of several novel miRNAs with multiple putative cellular targets [187]. Influenza virus infection also alters the expression of many lncRNAs. Most of the modified lncRNAs are involved in the cellular innate immune response, but the precise function needs to be further determined [188,199,200,201]. The IAV infection of A549 cells significantly altered the expression pattern of small nucleolar RNAs (snoRNAs). According to research by Zhuravlev et al., 66 snoRNAs were upregulated but 55 were downregulated upon IAV infection. The influenza viral regulation of ncRNAs is considered to be a complicated cellular event, requiring future exploration.

## 4. Conclusions and Perspectives

During influenza virus infection, complex interactions take place between viruses and their hosts [202,203,204,205]. The viruses are capable of positively regulating a large number of cellular proteins at different expression levels or biological activity status. Some of their inductions are due to the host’s antiviral response. For example, toll-like receptors (TLRs), retinoic acid-inducible gene-I-like receptors (RLRs), and their downstream signaling components are induced and/or activated during influenza virus infection, leading to the expression of host restriction factors, such as ISGs, to ultimately inhibit viral replication. In contrast, there are also many proteins upregulated, which appear to be directly induced by viral components to promote viral propagation.

As reviewed above, a considerable number of cellular proteins have been reported to be positively manipulated by influenza viruses during infection (Table 1). These upregulated proteins are involved in diverse signaling pathways, affecting multiple stages of the influenza virus life cycle, such as viral entry, transcription, replication, vRNP nuclear export, and virion assembly. This is reasonable because as acellular organisms, the viruses have to utilize host mechanisms and materials to fulfill their own activity and replication, which is usually achieved by hijacking supportive host factors. The induction of certain cellular proteins also contributes to the heightened pathogenicity and virulence of the viruses, especially for those highly pathogenic strains. Additionally, the upregulation of specific proteins is essential for the virus to effectively suppress the host immune response. The results should be validated using diverse experimental systems to determine whether the host factors are essential for or weakly supportive of the virus replication process. This will help to identify the host protein to target for developing potent antiviral drugs.

In addition to the proteins summarized in this article, there are also cellular proteins documented to be essential for the influenza virus life cycle without evidence showing the changes in their levels. For example, the non-structural protein NS2 regulates viral transcription, the nuclear export of vRNP, and virus budding through interacting with multiple cellular proteins such as CRM1, CHD3, NPC, and Nip98 [29,206,207,208,209,210]. However, the expression levels of these proteins seem to remain unchanged during infection. Interestingly, the artificial overexpression of these proteins under certain experimental conditions usually enhances influenza virus propagation, but the infection itself does not seem to alter the protein levels under natural conditions. There are several possibilities: (1) the basic expression levels of these proteins are sufficient for supporting the virus; (2) there may be some cellular feedback mechanisms that control the levels of these proteins, preventing them from being overexpressed. It is also possible that the authors did not specifically examine the expression of these proteins under their experimental settings.

Strategies for developing antiviral drugs can be largely categorized into two approaches: targeting the virus itself or the host cellular factors. Drugs targeting the virus itself have a drawback, which is the tendency to develop resistance. Novel therapeutic strategies may focus on targeting the cellular factors that are indispensable for the virus life cycle. Therefore, greater insight into the cellular proteins that benefit influenza virus replication will not only increase our understanding of virus–host interactions but also help to improve the design of new therapeutics against influenza.

## Figures and Tables

**Figure 1 ijms-26-03584-f001:**
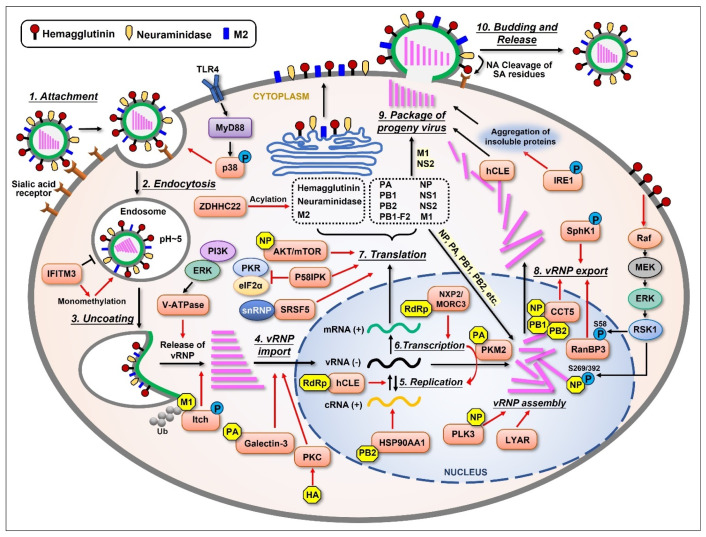
Schematic representation of the influenza virus life cycle and the cellular proteins that are upregulated by the virus. The influenza virus life cycle can be divided into several stages: (1) Virus attachment to the target cell; (2) Entry into the host cell; (3) Fusion of the endosomal membrane with the viral envelope, releasing viral ribonucleoprotein (vRNP); (4) Import of vRNPs into the nucleus; (5) Negative-sense viral RNA [vRNA(−)] is converted into positive-sense RNA [cRNA(+)] to serve as a template for the replication of progeny vRNA(−); (6) The virus also utilizes the host’s transcriptional machinery to generate its own mRNA(+); (7) The mRNAs translocate to the cytoplasm and are translated into at least 11 viral proteins. Three viral proteins found within the viral envelope, HA, NA, and M2, are transported to the cellular plasma membrane through ER-Golgi. Viral proteins such as PA, PB1, PB2, and NP are imported into the nucleus to form a progeny vRNP complex together with the newly synthesized vRNA(−); (8) The newly assembled vRNPs are then exported from the nucleus to the cytoplasm; (9) The newly generated viral components (proteins and vRNPs) are packaged into progeny virions and bud from the host cell; (10) During this step, the virus utilizes the host cell’s plasma membrane to form its own envelope, which contains viral HA, NA, and M2 on the surface. Lastly, viral NA must cleave the sialic acid residues from glycoproteins and glycolipids, helping viral particles release from the host cell. Several cellular proteins are shown to be upregulated (orange) by different viral components (yellow), thus displaying important supportive effects (red arrows) in most stages of the influenza virus life cycle.

**Figure 2 ijms-26-03584-f002:**
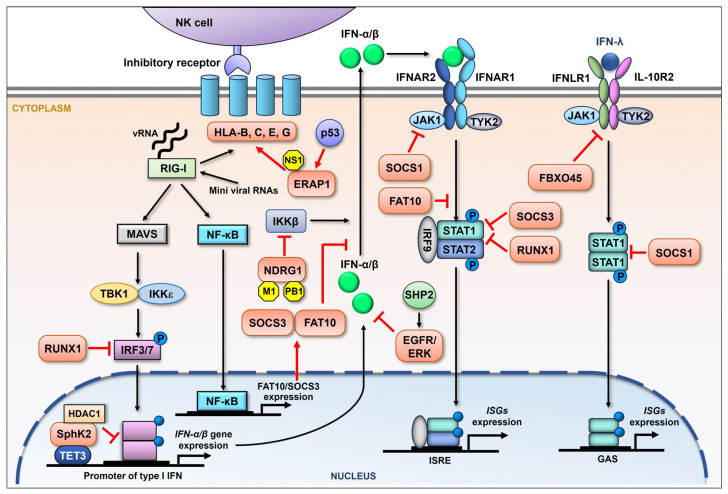
Schematic representation of the influenza viral suppression of the host immune response by upregulating cellular proteins. During influenza virus infection, RIG-I is primarily activated by the recognition of viral RNA (vRNA). The downstream adaptor, mitochondria-anchored protein MAVS, is then engaged, followed by the recruitment of IκB kinase ε (IKKε) and TANK-binding kinase 1 (TBK1), both of which are able to activate interferon regulatory factor (IRF) 3 and IRF7 by phosphorylation. Phosphorylated IRF3/7 then enters the nucleus, driving the production of IFN-β and variant IFN-α. After being secreted, IFN-I binds to the cognate receptor (IFNAR) and activates the JAK-STAT signaling pathway, involving the Janus kinase (JAK) family and the transcription factors STAT1/2. Phosphorylated STAT1 and STAT2 form a complex, leading to the induction of hundreds of ISGs that limit influenza viral replication. RIG-I signaling can also activate NF-κB, promoting the expression of a series of genes. Besides type I IFNs, there is type III IFN (IFN-λ) signaling, which is also proven to display antiviral activity by inducing the expression of type III IFN-stimulated genes. Influenza virus is reported to increase the expression of a batch of cellular proteins (orange) which regulate the NK cell and interferon-mediated innate immune response in many ways (red arrows and suppression symbols).

**Figure 3 ijms-26-03584-f003:**
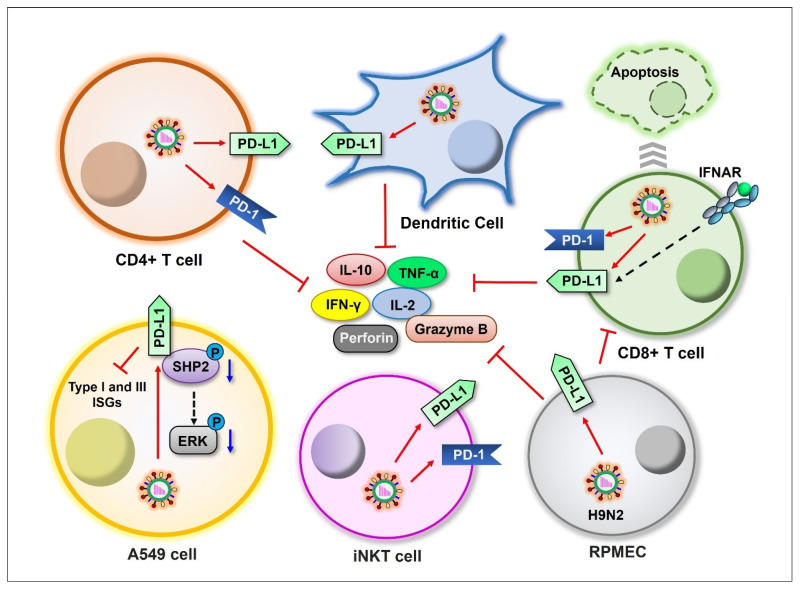
Schematic representation of influenza viral induction of PD-1/PD-L1 expression in different cell types. Influenza viruses induce the expression of PD-1 and/or PD-L1 on CD4+ T, CD8+ T, dendritic, iNKT, RPMEC, and A549 cells (red arrows), suppressing cytokine production. Blue arrows indicate downregulation.

**Figure 4 ijms-26-03584-f004:**
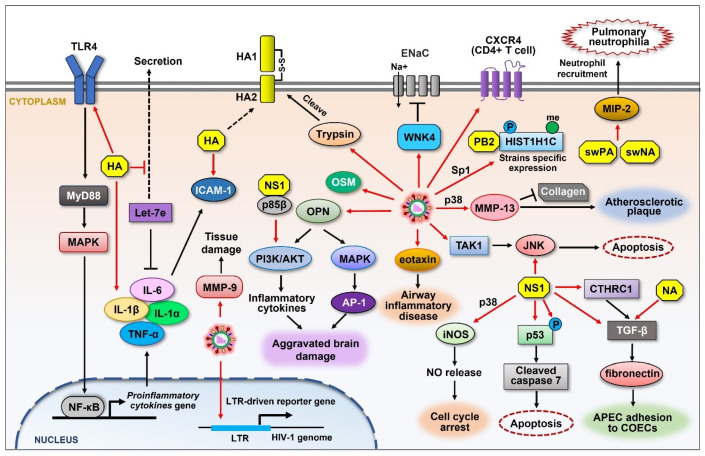
Schematic representation of the upregulation of cellular proteins that control influenza viral pathogenicity. Influenza viruses directly promote the expression of many cellular proteins (*red arrows*) that are involved in diverse signaling pathways. Viral HA induces the expression of ICAM-1 as well as proinflammatory cytokines such as IL-6, IL-1α, IL-1β, and TNF-α in a TLR4 signaling-dependent manner. Cellular trypsin is increased upon IAV infection, which cleaves HA0 into functional structures. IAV infection increases the expression of OPN, resulting in aggravated brain damage. IAV elevates the expression of eotaxin, which mediates airway inflammatory disease. IAV activates the expression of TAK1, leading to the activation of JNK. JNK is also activated by NS1, resulting in apoptosis. IAV induces the expression of WNK4 and inhibits ENaCs, impairing fluid transport in the respiratory system. The levels of CXCR4 on CD4+ T cells are increased during IAV infection, which affects HIV infection. IAV induces the expression of MMP-13 and MMP-9 and causes atherosclerotic plaque and tissue damage, respectively. H5N1 IAV induces the phosphorylation and methylation of HIST1H1C. Viral NS1 directly induces the expression or activation of several cellular proteins such as iNOS, p53 (p-p53), CTHRC1, and TGF-β, resulting in different cellular effects that benefit the viruses. Viral PA and NA from the swine influenza virus elevate MIP-2, causing pulmonary neutrophilia. Black arrows and graphics indicate regulatory effect that are not directly caused by viral component.

**Table 1 ijms-26-03584-t001:** Influenza-viral-directed upregulation of host cellular proteins.

Cellular Proteins	Effects of Cellular Proteins on Influenza Viruses	Biological Processes	Viral Component	References
p38 MAPK	Mediates virus entry in a TLR4- and MyD88-dependent way	Virus life cycle	ND	[33]
Itch	Regulates the entry-uncoating stage of influenza virus	Virus life cycle	ND	[34]
V-ATPase	Facilitates the release of vRNP into the cytoplasm by acidifying the endosomal interior	Virus life cycle	ND	[35,36]
PKC	Regulates influenza virus entry	Virus life cycle	HA	[37,38]
IFITM3-K88me1	Regulates viral membrane fusion to the endosomal membrane	Virus life cycle	HA	[39,40,41,42]
galectin-3	Increases the nuclear import of vRNP, viral transcription, and replication	Virus life cycle	PA	[43]
NXP2/MORC3	Plays important role in the transcription step of the virus life cycle	Virus life cycle	RdRp	[45,46,47]
PKM2	Transfers a phosphate group to PA and converts the function of viral RNA polymerase from transcriptase to replicase	Virus life cycle	RdRp	[48]
mTOR	Promotes the translation and protein synthesis of influenza virus	Virus life cycle	NP	[49,50,51]
HSP90AA1	Interacts with PB2 and increases viral RNA synthesis	Virus life cycle	PB2	[50,51]
P58IPK	Suppresses PKR and elF2α, resulting in enhanced translation of influenza viral mRNA	Virus life cycle	NP	[52,53,54,55]
SRSF5	Promotes viral M mRNA splicing and M2 production	Virus life cycle	ND	[56]
ZDHHC22	Mediates the acylation of influenza viral proteins	Virus life cycle	NS1	[59]
LYAR	Enhances vRNP assembly	Virus life cycle	vRNA	[60]
PLK3	Increases the phosphorylation and oligomerization of NP, thus promoting vRNP assembly	Virus life cycle	NP	[61]
RanBP3	Associates with CRM-1 and plays a crucial role in mediating vRNP export	Virus life cycle	ND	[62,63]
Raf	Promotes the nuclear export of vRNP	Virus life cycle	NP	[65,66]
SphK1	Regulates the synthesis of viral proteins and RNAs; mediates the CRM1-dependent nuclear export of vRNA	Virus life cycle	ND	[67]
CCT5	Interacts with viral NP, PB1, and PB2, mediating vRNP nuclear export	Virus life cycle	NP, PB1, PB2	[68]
hCLE/C14orf166	Stimulates both viral RNA polymerase and host Pol II; interacts with vRNP and incorporates into viral particles	Virus life cycle	RdRp	[73]
IRE1	Mediates the unfolded protein response and favors virion assembly by inducing the accumulation of insoluble protein aggregates	Virus life cycle	ND	[76]
FAT10	Facilitates viral replication by inhibiting type I IFN signaling	Immune evasion	vRNA	[80]
SOCS-3	Suppresses the type I IFN response by inhibiting STAT1 phosphorylation	Immune evasion	ND	[81]
SOCS-1	Disrupts the IFN-λ antiviral response by inhibiting STAT1 activation; mediates the ubiquitination and degradation of JAK1	Immune evasion	ND	[82,83]
FBXO45	Induces the ubiquitination and degradation of IFN-λ receptor IFNLR1	Immune evasion	ND	[84]
RUNX1	Hampers the expression of IRF3 and STAT1	Immune evasion	ND	[85]
SphK2	Associates with TET3 and HDAC1 and negatively regulates IFN-β transcription	Immune evasion	ND	[86,87,88]
EGFR	Regulates the type I and type III IFN-mediated antiviral response	Immune evasion	ND	[89,90]
NDRG1	Targets IKKβ and suppresses IFN-β production in a NF-κB-dependent way	Immune evasion	M1, PB1	[91]
HLA-G, -B, -C, -E	Binds to the inhibitory receptors of NK cells and helps influenza virus evade NK cells	Immune evasion	ND	[96,97,98,99]
ERAP1	Increases the surface expression of HLA in a p53-dependent manner	Immune evasion	NS1	[98]
PD1/PD-L1	Impairs T cell response against IAV by reducing cytokine expression and promoting the cell death of CD8+ T cells; reduces the expression of ISGs	Immune evasion	ND	[104,105,106,107,108,109,110,111,112,113]
IL-1α, IL-1β, IL-6	Responsible for the inflammatory effect caused by influenza virus	Viral pathogenicity	HA	[114]
ICAM-1	Plays an important role in pathological and inflammatory responses upon IAV infection	Viral pathogenicity	HA	[115]
WNK4	Promotes virus-induced reduction in ENaC activity, thus impairing the fluid transport in the airway	Viral pathogenicity	ND	[119]
MIP-2	Mediates neutrophil recruitment, resulting in substantial pulmonary neutrophilia	Viral pathogenicity	swNA, swPA	[123,124]
OPN	Results in aggravated brain damage and inflammation	Viral pathogenicity	ND	[128]
Trypsin	Potentiates brain vascular hyperpermeability and tissue damage; participates in the progression of myocarditis in severe influenza	Viral pathogenicity	ND	[129,130,131]
MMP-13	Decreases the level of collagen in the atherosclerotic plaque lesion, leading to the destabilization of vulnerable atherosclerotic plaques in the artery	Viral pathogenicity	ND	[134]
MMP-9	Plays major roles in the pathogenesis of severe IAV infection	Viral pathogenicity	ND	[135,136]
eotaxin	Participates in the pathogenesis of airway inflammatory disease caused by influenza	Viral pathogenicity	ND	[137]
OSM	Play roles in chronic rhinosinusitis with nasal polyps	Viral pathogenicity	ND	[138]
HIST1H1C	Positively regulates IAV H5N1 but negatively regulates the H1N1 virus	Viral pathogenicity	PB2	[143]
CXCR4	Contributes to HIV disease progression by modulating coreceptor availability	Viral pathogenicity	ND	[148]
LTR-driven reporter gene	Promotes HIV-1 transcription in CD4+ T cells	Viral pathogenicity	ND	[149]
p53	Induces mitochondrial dysfunction, contributing to H7N9/NS1-induced apoptosis	Cellular pathway	NS1	[154]
Cleaved caspase 7				
Cleaved PARP				
iNOS	Increases NO release in Neuro2a cells, inducing cell growth arrest and cellular senescence	Cellular pathway	NS1	[156]
PI3K	Supports viral propagation in multiple ways	Cellular pathway	NS1	[6,161,162]
JNK	Induces apoptosis; regulates vRNA and protein synthesis	Cellular pathway	NS1	[163,164,165,166,167]
TAK1	Responsible for IAV-induced JNK activation and apoptosis	Cellular pathway	ND	[165]
TGF-β	Induces apoptosis during influenza virus infection	Cellular pathway	NA	[168,169,170,171]
fibronectin	Induced by TGF-β signaling upon IAV infection, promoting *Escherichia coli* adhesion onto chicken oviduct epithelial cells	Cellular pathway	NS1	[172,173]
CTHRC1	Regulates TGF-β signaling and repairs vascular intimal injury	Cellular pathway	NS1	[174]
PRPF8	Enhances influenza viral replication	ND	NS1, PB1	[175,176]
TNKS1/2	Catalyzes enzymatic activities of the transferring of ADP-ribose motifs to target proteins	ND	ND	[183,184]

ND: not determined.

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
