# Peer review of "Positive Regulation of Cellular Proteins by Influenza Virus for Productive Infection"

_ijms, 2025, doi:10.3390/ijms26083584_

Round 1
Reviewer 1 Report
Comments and Suggestions for Authors
Cong et al. demonstrate that influenza viruses complete their infection cycle by exploiting host cell proteins and cellular pathways, with certain virus-upregulated cellular proteins acting as proviral factors essential for viral replication and pathogenicity. Investigating the mechanisms by which influenza viruses hijack cellular proteins could establish a theoretical framework for developing host factor-targeted therapeutics, ultimately enabling effective control of influenza.
The following are some comments and suggestions that are given to improve the manuscript:
Comment 1: The article repeatedly mentions that viral components (such as NS1, PB2, etc.) can upregulate host proteins, but the molecular mechanisms of upregulation are inadequately described. Please elaborate on how viral proteins regulate host gene expression, particularly addressing which level is most affected-transcription, translation, or protein stability.
Comment 2: Sections 3.1.2 and 3.1.3 have identical titles ("Viral entry, uncoating, and vRNP import"). This appears to be a formatting error. Please clarify if this is intentional and, if so, explain the rationale for having two sections with the same title.
Comment 3: Please describe the differences in host protein upregulation patterns among different influenza virus subtypes (H1N1, H3N2, H5N1, etc.) and discuss whether these differences correlate with variations in viral pathogenicity.
Comment 4: The regulatory role of non-coding RNAs (miRNAs, lncRNAs, etc.) in influenza virus infection is only briefly discussed in the manuscript. Please elaborate on their role in the upregulation of host proteins and provide information on the latest research developments in this area.
Comment 5: Please compare the strategies for upregulating host proteins between influenza viruses and other respiratory viruses (such as RSV, SARS-CoV-2), highlighting unique aspects of influenza mechanisms. Additionally, discuss how these distinctive features might inform the development of influenza-specific antiviral strategies.
Comment 6: Among the numerous upregulated proteins listed in Table 1, please identify which ones you consider essential for viral replication versus those that are merely supportive of the process. Discuss the significance of this distinction for developing targeted antiviral strategies.
Author Response
Comment 1: The article repeatedly mentions that viral components (such as NS1, PB2, etc.) can upregulate host proteins, but the molecular mechanisms of upregulation are inadequately described. Please elaborate on how viral proteins regulate host gene expression, particularly addressing which level is most affected-transcription, translation, or protein stability.
Response 1: We have previously referred the mechanisms for some of the upregulation and described the stages at which the cellular proteins were affected, e.g., line 174-175, line 208-209, line 256-257, etc. However, many research reports have shown that viral components upregulate cellular proteins (at least ultimately manifested as an increase in protein levels) where the exact mechanisms were not clearly defined in most studies. Therefore, it is difficult to elaborate on it if the increase was due to the regulation at the level of transcription, translation, or protein stability. Nevertheless, we have added more conclusions as much as we could regarding how the host proteins are upregulated. Modifications were made in line 169, line 178-179, line 181, line 197, etc.
Comment 2: Sections 3.1.2 and 3.1.3 have identical titles ("Viral entry, uncoating, and vRNP import"). This appears to be a formatting error. Please clarify if this is intentional and, if so, explain the rationale for having two sections with the same title.
Response 2: Section 3.1.3 has title “vRNP assembly and nuclear export.” However, we noted that both sections 3.1.1. and 3.1.2 had identical titles in the prior manuscript. We changed the title of 3.1.2 to “Viral replication, transcription, and translation”. We thank the reviewer for finding the error.
Comment 3: Please describe the differences in host protein upregulation patterns among different influenza virus subtypes (H1N1, H3N2, H5N1, etc.) and discuss whether these differences correlate with variations in viral pathogenicity.
Response 3: As discussed in line 581-583, it has been implied that the induction of certain host proteins contributes to the heightened pathogenicity and virulence of certain strains of IAV. For example, Zhang et al. have shown that HA from H7N9 strain increased the expression of several cytokines and miRNA let-7e. They believe that HA is likely an important pathological protein component of H7N9. However, little research has been done to directly compare the differential regulatory patterns among different IAV subtypes.
Comment 4: The regulatory role of non-coding RNAs (miRNAs, lncRNAs, etc.) in influenza virus infection is only briefly discussed in the manuscript. Please elaborate on their role in the upregulation of host proteins and provide information on the latest research developments in this area.
Response 4: We agree that recent studies have shown the regulatory effect of host ncRNAs on influenza virus. A large number of ncRNAs including microRNAs, lncRNAs, and others have been shown to be differentially expressed during IAV infection, and some of them appear to play key roles in influenza virus life cycle or host immune response. However, this review is focused on host proteins upregulated by viral proteins or as host defense. The detailed roles of regulatory ncRNAs during influenza have been summarized in other review articles. Due to the scope of this review, we prefer to keep the content as it is. Instead, we have added more recent references to this part to improve the quality of the article (line 550).
Comment 5: Please compare the strategies for upregulating host proteins between influenza viruses and other respiratory viruses (such as RSV, SARS-CoV-2), highlighting unique aspects of influenza mechanisms. Additionally, discuss how these distinctive features might inform the development of influenza-specific antiviral strategies.
Response 5: We agree that comparison between respiratory viruses such as SARS-CoV-2, RSV, and influenza virus would be a wonderful idea. However, we consider that this topic would form a separate review and is out of scope of this review. The editor suggested a minor revision, and the other reviewer evaluated that this is a good review with already over 200 references and described over 50 up-regulation of host cellular proteins during IAV infection. Further expansion of this review could diffuse the research summary. Thus, this review will remain to be focused on influenza virus.
Comment 6: Among the numerous upregulated proteins listed in Table 1, please identify which ones you consider essential for viral replication versus those that are merely supportive of the process. Discuss the significance of this distinction for developing targeted antiviral strategies.
Response 6: While we understand the reviewer’s comment, we like to maintain objective perspectives regarding the potency of many cellular proteins in regulating virus infection (essential vs. supportive). Investigators used their own experimental conditions. The results should be validated using diverse experimental systems over time to determine whether the host factors are essential for or weakly supportive of the virus replication process. This will help to identify the host protein to target for developing potent antiviral drugs in the future. We have now discussed this point in the Conclusions and Perspectives section (line 586-590).
Reviewer 2 Report
Comments and Suggestions for Authors
The authors reviewed host cell proteins up-regulated upon influenza A virus infection to provide a framework for inventing the host factor-targeted drugs to conquer influenza. The authors reviewed over 200 references and described over 50 up-regulation of host cellular proteins during IAV infection. This is a good review and balanced assessment of the cellular proteins regulation by influenza A virus. Specific comments follow.
Major points:
- Line 61: “IAV (H1N1)” should be “influenza A virus (IAV)”.
- Figure 1, 1.Attachment” Please explain why NA cleavage does not occur unlike 10.Budding and Release.
- Please consider to include; Protein turnover regulation is critical for influenza A virus infection. Huang Y, Urban C, Hubel P, Stukalov A, Pichlmair A. Cell Syst. 2024 Oct 16;15(10):911-929.e8. doi: 10.1016/j.cels.2024.09.004. Epub 2024 Oct 4. PMID: 39368468
Minor points:
- Line 42: Please spell out “NP”.
- Line 64: “(type I IFNs)” should be “(IFNs)”.
Author Response
The authors reviewed host cell proteins up-regulated upon influenza A virus infection to provide a framework for inventing the host factor-targeted drugs to conquer influenza. The authors reviewed over 200 references and described over 50 up-regulation of host cellular proteins during IAV infection. This is a good review and balanced assessment of the cellular proteins regulation by influenza A virus. Specific comments follow.
Response: We thank the reviewer very much for the positive evaluation.
Major points:
- Line 61: “IAV (H1N1)” should be “influenza A virus (IAV)”.
- Figure 1, 1.Attachment” Please explain why NA cleavage does not occur unlike 10.Budding and Release.
- Please consider to include; Protein turnover regulation is critical for influenza A virus infection. Huang Y, Urban C, Hubel P, Stukalov A, Pichlmair A. Cell Syst. 2024 Oct 16;15(10):911-929.e8. doi: 10.1016/j.cels.2024.09.004. Epub 2024 Oct 4. PMID: 39368468
Response:
- We have changed it as suggested. We thank the reviewer for the correction.
- There have been reports about the role of NA cleavage activity in viral attachment and entry processes as well as the viral releases as reviewed in Yang et al., Rev Med Virol. 2016. Since the suggested role of NS in viral attachment and entry is diverse, we refrain from describing the details in this review.
- The listed reference has been added.
Minor points:
- Line 42: Please spell out “NP”.
- Line 64: “(type I IFNs)” should be “(IFNs)”.
Response:
All suggested modifications (Minor points 1 and 2) have been made accordingly.